# Recovering Microscopic Images in Material Science Documents by Image Inpainting

## Taeyun Kim and Byung Chul Yeo *

Department of Energy Resources Engineering, Pukyong National University, Busan 48513, Republic of Korea
* Correspondence: byeo@pknu.ac.kr

**Abstract:** Microscopic images in material science documents have increased in number due to the growth and common use of electron microscopy instruments. Through the use of data mining techniques, they are easily accessible and can be obtained from documents published online. As data-driven approaches are becoming increasingly common in the material science field, massively acquired experimental images through microscopy play important roles in terms of developing an artificial intelligence (AI) model for the purposes of automatically diagnosing crucial material structures. However, irrelevant objects (e.g., letters, scale bars, and arrows) that are often present inside original microscopic photos should be removed for the purposes of improving the AI models. To avoid the issue above, we applied four image inpainting algorithms (i.e., shift-net, global and local, contextual attention, and gated convolution) to a learning approach, with the aim of recovering microscopic images in journal papers. We estimated the structural similarity index measure (SSIM) and $\ell_1/\ell_2$ errors, which are often used as measures of image quality. Lastly, we observed that gated convolution possessed the best performance for inpainting the microscopic images.

**Keywords:** microscopic images; material science literature; image inpainting

## 1. Introduction

Microscopic images in material science documents (e.g., books, articles, and reports) are generally used to display key structures in a target material [1]. The images are produced by various types of microscopic imaging tools, such as optical microscopy, transmission electron microscopy (TEM), scanning TEM (STEM), scanning electron microscopy (SEM), etc. [2–6]. Due to the advancement and widespread use of such microscopy instruments, micro-/nano-scale material images have become more sophisticated and prevalent in material science documents.

In the material science field, there is a massive volume of online documents that have been published. In addition, digitalized material datasets, e.g., texts, graphs, and images, are included within them. Due to open access movements with respect to scientific publishing, a huge number of digitized material images are freely available to be downloaded via the processes of web crawling or scraping, which are data mining techniques [7]. If one learns even a small degree of python programming regarding the processes of data scraping, then the material images through microscopy are easily obtained and utilized.

Since the advent of the data-driven approach, these material datasets are useful in terms of feeding artificial intelligence (AI) models that enable them to efficiently find explicit material properties [8]. In particular, AI-driven analysis on such material images can accelerate the process of diagnosing real space geometrical information with a higher accuracy instead of the analysis conducted by humans [9,10]. To guarantee a high-quality AI model, it is necessary that the training dataset consists of abundant and actual microscopic images [11]. Following this, the diverse experimental images obtained via microscopy and which are stored in the material science documents are applicable to the training dataset. However, it gives rise to two main issues.

Firstly, when we inspect the microscopic images acquired in the documents, several marks, such as letters, scale bars, and lines/arrows, are added into the raw microscopic photos for the purposes of either expressing image information or for highlighting sections [12–14]. For example, letters explain ordering in the figure set, the scale bar shows imaging resolution, and lines/arrows indicate remarkable objects. Therefore, one needs to remake the published images to remove the marks.

Indeed, image inpainting techniques have been developed in the computer vision field for the purposes of filling in the adequate information required in the missing parts of target images [15]. Moreover, in recent years, they were designed by not only rule-based models but also deep learning-related models in order to achieve a good performance in terms of accuracy and speed, albeit these models were focused only on retrieving original figures regarding people and landscapes [16–26]. Thus, we applied various image inpainting models with the aim of recovering microscopic images in material science documents, specifically in regard to solving the problem detailed above.

In this study, we contribute the main things, as follows: (1) we proposed new methods regarding the preprocessing of inputs with statistical and threshold-based masks, specifically with respect to the latest image inpainting methods for recovering microscopic images in material science documents; and (2) we compared the performances of the various image inpainting models in terms of their restorative performance factors, which were related to the similarity with the ground truth of the images.

The remainder of this paper is organized as follows: Section 2 introduces the related work and backgrounds of the microscopic images in material science documents and the image inpainting techniques that were utilized in the computer vision field. In Section 3, we describe the important details in our works of statistical and threshold-based masking, as well as detail a comparison of the image similarity performances between the various models. Finally, in Section 4, we conclude our research work and discuss the implications of our model.

## 2. Related Works and Backgrounds

### 2.1. Microscopic Images in Material Science Documents

To allow readers to more easily understand each microscopic image in material science documents, two main additives are used. These additives are an alphabet character and a scale bar. The first additive, i.e., the alphabet character, is used for specifying its turn among multiple images in a figure set. The second additive, i.e., the scale bar (specifically, the scale ruler) is used for visually indicating distance and size within the image.

In Figure 1, the characters and scale bars that are disclosed on the TEM, STEM, and SEM images are inserted into the reference papers [12–14]. In addition, they were placed at the four corners of the images: top-left, top-right, bottom-left, and bottom-right. In this paper, our objective is to replace, specifically in the microscopic images within the material science papers, the additives by their most plausible replacement pixels as one example of image inpainting tasks.

### 2.2. Image Inpainting Methods

Image inpainting methods in the field of computer vision are mainly classified into two categories: non-learning approaches and deep learning approaches [15]. Moreover, non-learning approaches are further divided into patch-based types and diffusion-based types, and deep learning approaches are further divided into convolutional neural network (CNN)-based types and generative adversarial network (GAN)-based types. The list of these approaches from the past to the current day, in terms of image inpainting methods, is presented in Table 1.

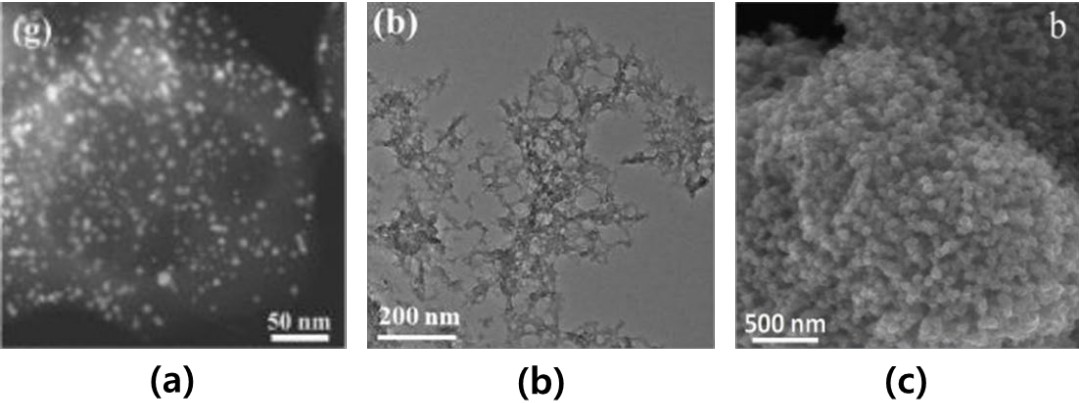

**Figure 1.** Examples of microscopic images in material science documents: (**a**) represents the STEM image of Au nanoparticles. reprinted with permission from Ref. [12]. Copyright (2020) Elsevier; (**b**) represents the TEM image of $Co_8Fe_2$-MOF. reprinted with permission from Ref. [13]. Copyright (2021) Elsevier; and (**c**) represents the SEM image of S/PCMSs composites. reprinted with permission from Ref. [14]. Copyright (2019) Elsevier.

**Table 1.** List of various image inpainting methods in the field of computer vision.

| Category | Type | Method | Dataset |
|---|---|---|---|
| Non-learning approach | Patch-based | Simakov et al. [16] | - |
| | | Bertalmio et al. [17] | - |
| | | Criminisi et al. [18] | - |
| | | PatchMatch algorithm [19] | - |
| | Diffusion-based | Baertalmio et al. [20] | - |
| | | Ballester et al. [21] | - |
| | | Levin et al. [22] | - |
| Deep-learning approach | CNN-based | Shift-Net [23] | Places [27], Paris street View [28] |
| | GAN-based | Global and Local [24] | Places2 [27], ImageNet [29], CMP FAcade [30] |
| | | Contextual Attention [25] | Places2 [27], ImageNet [29], CelebA [31], CelebA-HQ [32], DTD [33] |
| | | Gated convolution [26] | Places2 [27] |

Among the two image inpainting approaches, the non-learning-based approach is known as the more traditional method; this method is often used to copy the most similar pixel information that is adjacent to the target areas and that correspond to either damaged or missing regions in the image [16–22]. On the other hand, the patch-based method in the non-learning approach is designed to locate the patches that are most similar to the target area by using random sampling and propagation instead of thoroughly searching the entire image. For instance, Barnes et al. proposed the PatchMatch algorithm, which operates by densely matching the patches between two images [19]. In addition, it is suitable for use with various images, as it is able to handle the desired texture and color changes. Diffusion-based methods in the non-learning approach operate by reconstructing target regions, which is achieved by analyzing the entire image and then gradually diffusing the

information (e.g., color, texture, and shape of the image) of pixels that are closest to the target region. For instance, Bertalmio et al. proposed a diffusion-based method that is based on partial differential equations (PDE), which allows the propagation of gradient information from the known regions to the target regions for the purposes of filling in the missing pixels [20]. In addition, it is also suitable for large-scale image inpainting tasks, as it is easy to implement and can be easily parallelized. However, this approach has two limitations: (1) it is useless for dealing with complex pixel information around large target areas, and (2) it requires a high computational cost.

In contrast to the non-learning approach, the deep learning approach can learn from and extract the complex features of large target areas in images with a more accurate and faster prediction. In Figure 2a, CNN-based methods basically follow an encoder–decoder network architecture [34–36]. Here, the encoder transforms inputs into a state within a latent space, whereby the decoder reconstructs the compressed outputs from the encoder [37]. For instance, Pathak et al. proposed a context encoder network that utilizes both L2 reconstruction loss and adversarial loss in order to improve visual quality [37]; however, this method is not suitable for high-resolution images. Yan et al. proposed a shift-net algorithm, which preserves the features of an image by introducing shift connections in a specific layer, thereby allowing the generation of high-resolution images from low-resolution inputs [23]. However, they still encountered difficulties in terms of not preserving details that were too sophisticated and thus only producing visually convincing results.

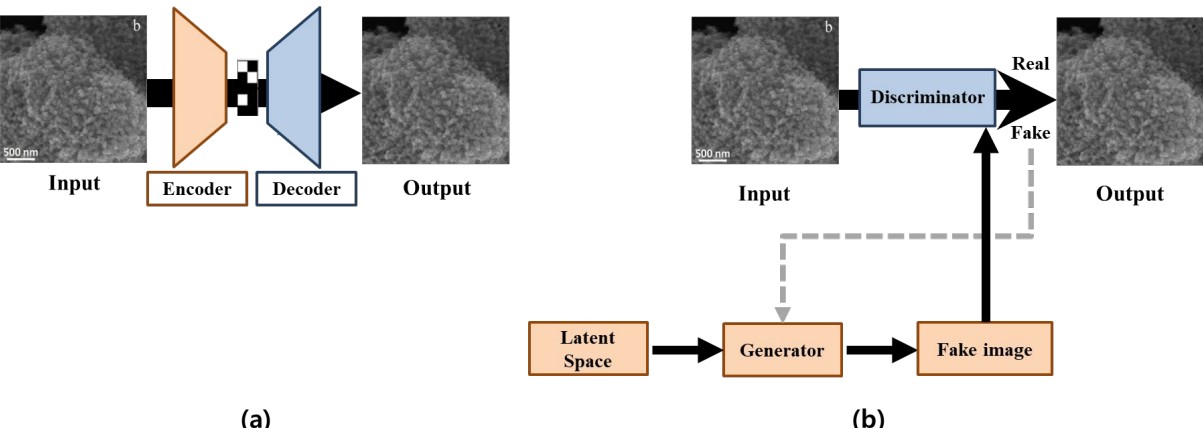

**(a)**                                                                    **(b)**

**Figure 2.** Architectures of deep learning methods (In the figures, input images are the example of the microscopic images in the reprinted with permission from Ref. [14]. Copyright (2019) Elsevier.): (**a**) represents the CNN-based method and (**b**) represents the GAN-based method.

In recent years, a GAN technique has appeared due to the advancement of machine learning algorithms. In Figure 2b, GAN-based architectures consist of generator and discriminator networks. In detail, the generator generates virtual images by using a feature map in the latent space, whereby the discriminator distinguishes the filled images from the real images [38]. For example, Iizuka et al. proposed the GAN-based method that uses a three-step process, which involved a completion network, a global context discriminator, and a local context discriminator in order to generate high-quality results for large areas of missing parts [24]. Additionally, Yu et al. proposed a generative network that uses a context attention module in order to maintain consistency with the surrounding context [25]. However, this method can still produce unrealistic results, such as a distorted structure when using free-form masks. In order to overcome the challenge, Yu et al. proposed a gated convolution to improve the color consistency and inpainting quality of free-form masks [26]. This method applied a gated convolution to each spatial location in all layers in order to solve the color mismatch and blurriness issues of vanilla convolutions.

In this study, we focused on deep-learning-based methods because they are superior to the other methods due to their lower computing costs and higher prediction performance

up to now. In particular, we employed one CNN-based method and three GAN-based methods, which are shift-net, global and local, contextual attention, and gated convolution. We used the same architecture in all of the models proposed in the references [23–26].

## 3. Results

### 3.1. Data Preparation

In order to train all the deep learning-based methods for image inpainting, we required a large dataset of the microscopy images in the material science documents, as well as the corresponding originals. We can utilize text mining and natural language processing in the material science field, and many material images in their respective journals can be obtained for oneself. Therefore, we extracted 1100 microscopy images from 129 material science papers, which were downloaded from Elsevier journals. Then, we obtained the original images by using, by hand, the content-aware fill technique in Photoshop [39]. Furthermore, in this paper, we considered that the original image obtained via microscopy corresponds to a ground truth. Moreover, the training and testing images of the whole dataset came to a total of 880 images (80%) and 220 images (20%), respectively. The input images were used at a size of 256-by-256. Additionally, the hardware specification to run the image inpainting methods was an Intel(R) Xeon(R) Silver 4214R CPU @ 2.40 GHz (Intel Corporation, Santa Clara, CA, USA) and an NVIDIA GeForce RTX 3090 GPU (Nvidia Corporation, Santa Clara, CA, USA).

### 3.2. Generating Inputs Using Masks

When we applied the image inpainting methods, the inputs should have been pre-processed as the microscopic images in the material science documents were convolved with a mask. Then, the mask was filled in either black or white colored pixel values for specifying normal or undesired areas. Generally, this would be a binary image and should have the same size as the input image [24,26]. In the mask image, the pixels that should be preserved are those which are black, and the pixels in the undesired areas are in white [15]. For training the model, we used a random mask [23]. Next, we used the mask that covers the two additives, i.e., the alphabet characters and scale bars, on the images for testing the model. There was, initially, a lack of information (e.g., type, location, and length) for the additives. However, the masks were then generated by two ways: statistical masking and threshold-based masking.

Firstly, we investigated the statistical positions and sizes of the two additives (the alphabet characters and scale bars) on the four corners in the 1100 given images. In Figure 3a, we observed that the averaged positions of the alphabet characters in the top-left and top-right were (23, 233) and (225, 228), respectively, whereas the averaged positions of the scale bars in the bottom-left and bottom-right were (41, 20) and (212, 26), respectively. In order to measure the average size of the alphabet characters and the scale bars, we estimated a cumulative distribution function (CDF) with respect to the ratio of their sizes and their probabilities, as shown in Figure 3b. Although all the additives in the four regions were only the characters and the scale bars, the different average sizes of the additives in the four regions were observed. The characters appeared at the top, but the scale bars appeared at the bottom. Since the sizes of characters and scale bars were different, the average sizes of the additives between the top and bottom were different. Furthermore, the average sizes of the additives among the four regions can be different because of the personal preference of the authors. For instance, one used the additive with parenthesis as like "(b)" (see Figure 1b), and the other one used the additive without parenthesis as like "b" (see Figure 1c). Then, we observed that the maximum sizes of the 80% majority additives in the top-left, top-right, bottom-left, and bottom-right were 36 (pixels) × 36 (pixels), 41 (pixels) × 41 (pixels), 49 (pixels) × 49 (pixels), and 51 (pixels) × 51 (pixels), respectively. Therefore, we decided on the statistical mask that covers the four squares at (23, 233), (225, 228), (41, 20), and (212, 26), respectively.

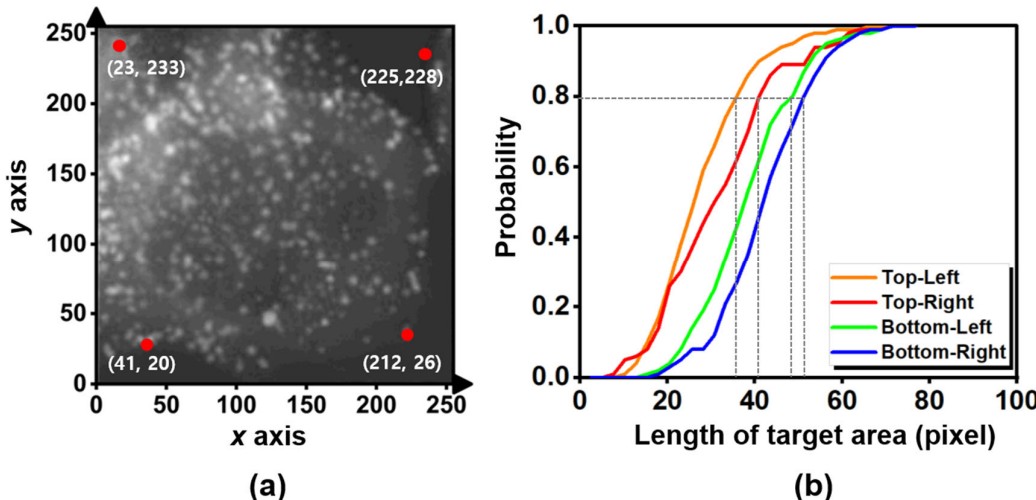

**Figure 3.** Statistical results of the two additives: (**a**) represents the averaged positions (red dots) of the alphabet characters and scale bars regarding the example of the microscopic images (reprinted with permission from Ref. [12]. Copyright (2020) Elsevier) and (**b**) represents the cumulative probability distribution function, according to the length of the target area (pixel) of the target regions comprising the additives.

Next, we designed an algorithm that generates a threshold-based mask. When we investigated 1100 images, we checked that the microscopic images mostly took on extremely white-colored additives against a dark background. Therefore, we defined the threshold-based mask to include white pixels of a higher threshold as the additives, as well as black pixels of a lower threshold for the others. For distinguishing the pixels of the additives, as well as the others, we used a threshold value of 200 in gray scale. In Figure 4a, this can be black on the additive, but white around the additive. When we used only a threshold value in order to distinguish the pixels of these, the additive filled the black pixels remains as a non-removable object, as can be seen in Figure 4b. Therefore, we added the function that the pixels close to the white pixels possess a white color in the algorithm. We observed that the regions of the additives can possess white, and the regions of the others can possess black with respect to the result of the algorithm (see Figure 4c). The threshold-based mask code that we implemented is available at https://github.com/hmnd1257/threshold-mask (accessed on 16 February 2023).

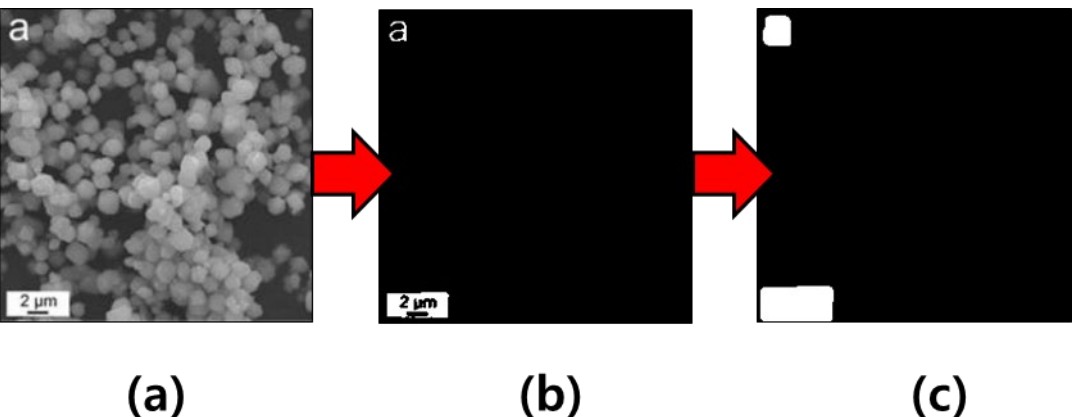

**Figure 4.** Processing sequence in regard to generating a threshold-based mask from the example of a microscopic image (reprinted with permission from Ref. [40]. Copyright (2021) Elsevier): (**a**) represents the input image; (**b**) represents the intermediate image in the context of threshold-based masking; and (**c**) represents the final image.

### 3.3. Comparisons of Masks and Models

In the first experiment, we confirmed the effects of two masks with respect to the statistical mask and the threshold-based mask. We tested whether the input that is used by the statistical masks via the use of shift-net was the most popular deep learning method. The additives of the microscopic images in the material science documents appeared mainly in the four corners (i.e., top-left, bottom-left, top-right, and bottom-right), but they often appear in other areas (i.e., the center). Since the statistical mask has fixed four corner areas, it is difficult to become flexible masking. In Figure 5b, in the case of the statistical mask, we observed good results in all the additive regions that were top-left and bottom-left, but the center region was not restored at all. However, in the case of the threshold-based mask, Figure 5c shows that we observed better results with respect to restoring all the additives inside the image. In particular, the results in the area around the C label in Figure 5b,c look different. The statistical mask used a fixed square mask that includes the area around the C label as well as the C label, but the threshold-based mask contains only pixels belonging to the C label. It means that the statistical mask is relatively larger than the threshold-based masks in the same situation. Then, the statistical mask should restore the area around the C label, and the restoring noise on the area can be inserted. Therefore, the result in the area around the additives by using the threshold-based masks can be better than the result in the area around the C label by using the statistical masks.

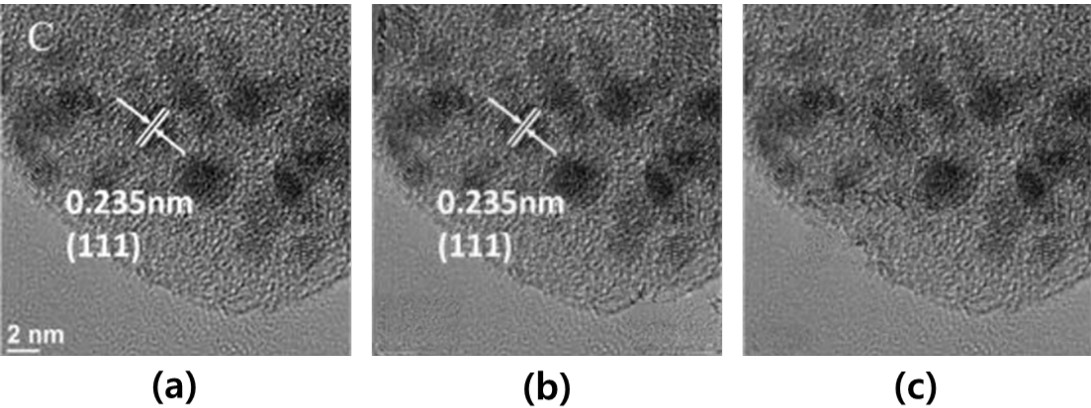

(a)　　　　　　　　　　　(b)　　　　　　　　　　　(c)

**Figure 5.** Comparison of the results, which was achieved by using a statistical mask and a threshold mask from the example of microscopic image (reprinted with permission from Ref. [41]. Copyright (2019) Elsevier): (**a**) represents input image; (**b**) represents output image using a statistical mask; and (**c**) represents the output image when using a threshold-based mask.

In the second experiment, we confirmed the effects of the four latest deep learning methods (shift-net, global and local, contextual attention, and gated convolution) for the purposes of image inpainting in our datasets of microscopy images. Through the first experiment, we tested the input used by the threshold-based masks via the use of the four latest deep learning methods. Then, we conducted qualitative evaluations of their results. Figure 6 shows three examples of the inputs and outputs that were obtained with the four different methods for qualitative comparison. It was found that the results of shift-net, global and local, and contextual attention were unsatisfactory because of blurriness in relation to the position of the characters and scale bar, as shown in Figure 6b–d. However, the gated convolution achieved better results than the other methods for all the examples, as shown in Figure 6e.

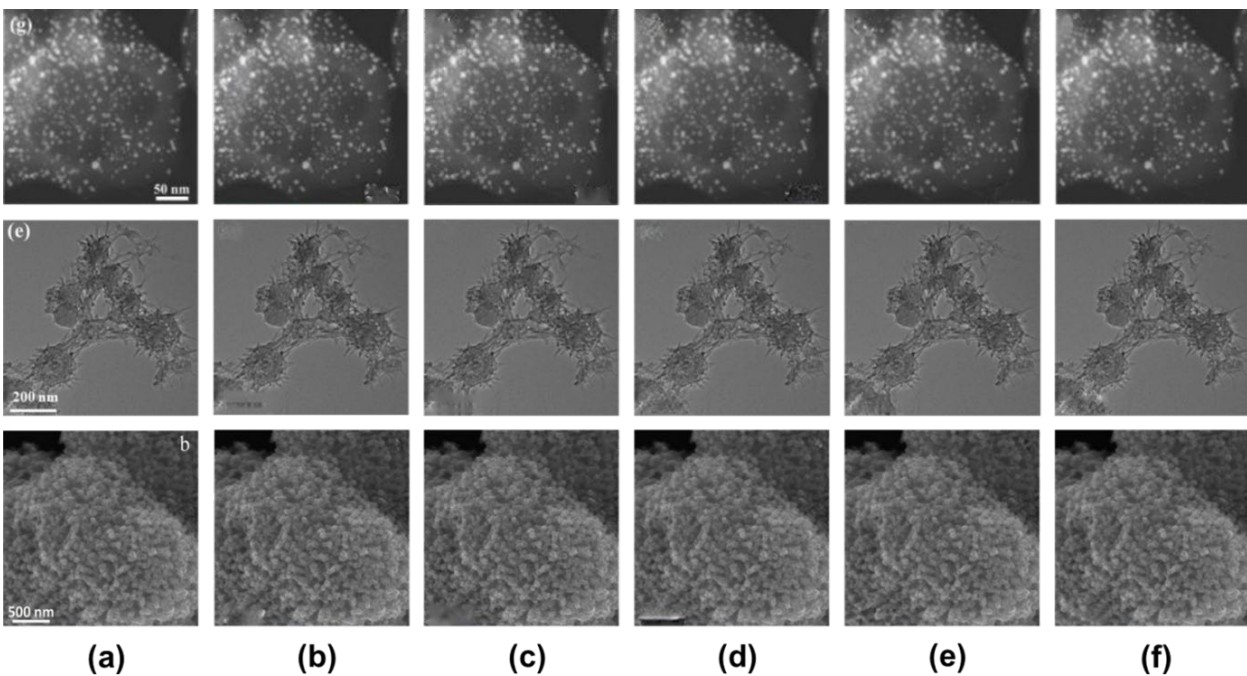

**Figure 6.** Comparison of the results of various image inpainting methods from the three examples of microscopic images (from top to bottom. reprinted with permission from Ref. [12]. Copyright (2020) Elsevier, from Ref. [13]. Copyright (2021) Elsevier, and from Ref. [14]. Copyright (2019) Elsevier): (**a**) represents the input image; (**b**) represents the output image of shift-net; (**c**) represents the output image of global and local; (**d**) represents the output image of contextual attention; (**e**) represents the output image of gated convolution; and (**f**) represents the ground truth.

Next, we conducted quantitative evaluations of their results. Furthermore, we used three evaluation metrics, the structural similarity index measure (SSIM) and $\ell_1/\ell_2$ errors [26,42]. Then, we tested the images in order to measure the three metrics for the four methods and then calculated the averages and standard deviations of each method. Then, the additive's area in the entire figure is usually small. Since all the pixels except restoring area in the outputs are the same as the ones in the originals, the metrics of the whole images with respect to none or using methods are nearly high score and their difference could be significantly trivial. To show the difference in the results between with and without the methods, we calculated the metrics of local regions where the characters and scale bars are closely packed. In Table 2, we observed that a gate convolution possessed the best performance measures of SSIM and $\ell_1/\ell_2$ errors.

**Table 2.** Averages and standard deviations in parenthesis of performance metrics (SSIM, $\ell_1$ and $\ell_2$ errors) for the image similarity results.

| Method | SSIM | $\ell_1$ Error | $\ell_2$ Error |
|---|---|---|---|
| None | 0.29 | 7.85% | 3.94% |
| Shift-Net | 0.80 (0.09) | 4.40% (1.91) | 1.99% (1.08) |
| Global and Local | 0.82 (0.09) | 4.24% (2.03) | 1.85% (0.79) |
| Contextual Attention | 0.89 (0.09) | 3.97% (1.72) | 1.79% (0.79) |
| Gated convolution | 0.94 (0.08) | 3.22% (1.69) | 1.57% (0.83) |

## 4. Conclusions

In this paper, we presented a practical use of image inpainting methods for recovering microscopic images in material science documents. In particular, we proposed and compared new methods of preprocessing the inputs with statistical and threshold-based masks in regard to the image inpainting methods. The image inpainting methods that we

used were four models that have appeared in the computer vision field, which are shift-net, global and local, contextual attention, and gated convolution. The gated convolution model showed the best SSIM and $\ell_1/\ell_2$ errors, which are the measures of the image similarity between the output and the ground truth. However, this work only shows microscopic images, but it must be said that the figures in the documents are diverse. In future works, we will extend our work to develop an advanced image inpainting method that robustly and efficiently recovers all the images comprising other figures, as well as the material images obtained via microscopy in the documents.

**Author Contributions:** T.K.: methodology, software, validation, analysis, writing; B.C.Y.: writing-review and editing. All authors have read and agreed to the published version of the manuscript.

**Funding:** This research was supported by the Nano & Material Technology Development Program through the National Research Foundation of Korea (NRF) funded by the Ministry of Science and ICT (2021M3A7C2090586).

**Institutional Review Board Statement:** Not applicable.

**Informed Consent Statement:** Not applicable.

**Data Availability Statement:** The data will be shared upon reasonable request.

**Conflicts of Interest:** The authors declare no conflict of interest.

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
