# Peer review of "Recovering Microscopic Images in Material Science Documents by Image Inpainting"

_applsci, doi:10.3390/app13064071_

Round 1

Reviewer 1 Report

Dear Editor, dear authors

I read the manuscript entitled “Recovering Microscopic Images in Material Science Documents by Image Inpainting” written by T. Kim and C. Yeo

The paper deals with the use and collection, by data mining or scraping, of microscopy images useful for the training of AI. The authors employed 4 different algorithms of image inpainting to recover microscopy images from published papers with the aims of tackling two main problems:

1) the microscopy images are usually contains additional unwanted marks such as scale bars, captions, sublabels etc

2) the recovered images are usually covered by copyright, thus unusable in a fine way

The authors explained in a step-by-step way the approach they performed when they produced the masks and tested the algorithms. They also explain the strengths and weakness of different masks and method used

As a whole, the paper plan is well described and simple. The results are well presented and the figures are clear and properly fitted the text. The discussion is exhaustive and the English is understandable.

In light of these I believe the paper can be published with minor revision.

My comments are listed below:

- A curiosity on Page 6: the authors wrote that “we observed that the maximum sizes of the 80% majority additives in the top-left, top-right, bottom-left, and bottom-right were 36 (pixels) × 36 (pixels), 41 (pixels) × 41 (pixels), 49 (pixels) × 49 (pixels), 51 (pixels) × 51 (pixels), respectively”

Is not clear to me if there is a reason for different average size observed for the additives placed in different corners? Maybe they are different kind of additives?

- In figure 5, the reconstruction of the microscopy images in the area with the C label reported in panels (b) and (c) are clearly different. Please, comment

- One exception to the proper English is reported below (from page 8). Please make more clear the following sentence:

“To show the difference of the results between with and without method extremely, we calculated the metrics…” 

- page 8-9: I don’t’ know if the paper is in a final template, but it would be better to shift table 3, as the titles are on page 8 while all the data are on page 9. 

Reviewer 2 Report

In the article, the authors apply artificial intelligence (AI) models for inpainting in order to eliminate labels from electron microscopy images.  Proper  inpainting techniques can be important for automatic analysis of the images extracted from the scientific literature.  Also, they can be useful for the preparation of illustration in reviews and educational projects. The authors compare four  deep learning based methods and conclude that the gated convolution model gives the best result. In general, I recommend accepting the article for publication, but there are a few comments. 1. The statement in the abstract "...their utilization under fair use would not apply because they are copyrighted..."  and the idea of remaking the published images to avoid copyright restriction are not clear.  In my opinion, the use of the published images for the training of an AI model in a non-commercial scientific project is fair use, and permission is not required. Thus, I recommend to exclude the mention of the copyright issue from the article. 2. If Table 3 shows the averages of evaluation metrics, then what are the deviations from these averages for different images? 3. Page 8,  line 6 from the bottom: should be three metrics instead of two.
